# Experimental Investigation of the Dielectric Constants of Thin Noble Metallic Films Using a Surface Plasmon Resonance Sensor

**DOI:** 10.3390/s20051505

**Published:** 2020-03-09

**Authors:** Longbiao Tao, Shuo Deng, Hongyun Gao, Haifei Lv, Xiaoyan Wen, Min Li

**Affiliations:** Department of Physics, Wuhan University of Technology, Wuhan 430070, China; taolongbiao555@gmail.com (L.T.); dengshuo1990@whut.edu.cn (S.D.); haifeilv@whut.edu.cn (H.L.); wenxy@whut.edu.cn (X.W.); minli@whut.edu.cn (M.L.)

**Keywords:** surface plasmon resonance, dielectric constants, visible and near infrared, multi-wavelength method

## Abstract

Gold and silver have an extremely low refractive index value of about 0.04 in the visible to near infrared (NIR) regions, and this induces a relative error of about 50% in refractive index measurements. This can lead to a large uncertainty in the imaginary part of the dielectric constants. A large difference exists between the experimental results and the classic models. The surface plasmon resonance (SPR) sensors, which use tens of nanometer thick noble metal film as the sensing layer, show ultra-high sensitivity (reaching 10^−8^ RIU) in this spectral range. As the spectral sensitivity and amplitude of SPR curves depend on the thickness and the dielectric constant of the sensing layer, we obtained high precision optical constants of the noble metal film using a multi-wavelength angle-modulated SPR sensing technology. The dielectric constant inferred from the parameters of the SPR curves, rather than from the refractive index and absorption ratio of noble metals, introduced a relative error within 10% of the resonance angle measurement. The measurement results demonstrate that the dielectric constants of gold and silver nano-films are more consistent with the widely used experimental results than with the classical theoretical model and always fall in the upper half of the imaginary part of the uncertainty range in the spectra of 500–900 nm.

## 1. Introduction

The dielectric constants of noble metals are critical in the design of thin film and nanoscale devices, in particular in photonic, plasmonic structure, and nanoparticle arrays [1]. Drude proposed a free electron model to explain the transport properties of electrons in metallic materials. However, classical mechanics cannot correctly describe the motion of microscopic particles. The Lorentz–Drude (L-D) model introduced a damping resonance term to ameliorate the excess electrons in the Drude model [2]. Since the Drude era, many modified Drude models were proposed for the precise estimation of the dielectric constants of various metals [3].

The measurement precision of the dielectric constant of noble metals is constantly being improved. Among them, Johnson and Christy [4,5,6] produced the most notable results, measuring the dielectric constants of copper, gold, and silver in the spectral range from 0.5 to 6.5 eV in the transmission, reflection, and 60° p-polarized modes. With knowledge of the exact film thickness, it is possible to measure the refractive index (*n*) and absorption ratio (*k*) of the whole spectrum using only two of the three measurements. As the imaginary part is calculated from the relation: $\varepsilon_{1}^{\prime \prime }=2nk$, the inherent error of the refractive index and absorption ratio in the experiments directly affect the results. However, the generation of current in plasma from free electrons causes the refractive index to be much smaller than one in this spectral region, and the error value is on the order of the magnitude, which results in a 50% relative error and a greater impact on the dielectric constant, and this is the same in similar optical experiments. Stahrenberg [7] and Baber [8] continued using Christy’s method with the exposure of the metal layer to air and with improved surface roughness, but kept the error introduction mechanism. Mcpeak [9] indicated that different film preparation processes are major contributors to the differences of experimental results. Neither the transmission reflection method nor ellipsometry reduces the uncertainty of the imaginary part of the dielectric constant in the visible and near infrared (NIR) regions. This phenomenon is clearly demonstrated in the results of Johnson and Christy [4].

As the spectral sensitivity and amplitude of surface plasmon resonance (SPR) curves depend on the thickness and dielectric constant of the sensing layer, SPR results are a better choice for obtaining the unique dielectric constant of silver or gold at a certain wavelength. After Lopez-Rios and Vuye [10] first proposed an SPR measurement method, Chen [11] and Innes [12] reported the dielectric constants of silver and gold nanofilms with SPR measurements and proved the validity of the experiment. However, their experiments were limited by the wavelength of the laser. They could not obtain continuous data to analyze the effects of the errors.

Liu et al. [13] proved the influence of the line width and the divergence angle of the light source. Wei et al. analyzed the effect of temperature [14]. Rehmanh [15] and Reale [16] gave the effects of thickness for both theory and experiments. Nash measured the dielectric function of silver through the excitation of surface plasmon-polaritons on a silver-covered silica grating [17]. Cao et al. [18,19] improved the SPR measurement method. An extended attenuated total reflection produces the dielectric constant measurement result in a single measurement. However, these structural improvements cannot effectively reduce the error in the optical measurement process.

Based on a typical prism–metal–air three-layer Kretschmann structure, we took advantage of a supercontinuum laser source to carry a set of multi-wavelength measurements in the visible to NIR range. Section 2 presents the theoretical model of dielectric constant derivation from the SPR parameters. Section 3 illustrates the experimental results, and Section 4 discusses, in detail, the effects of the light source, temperature, and film thickness on measurement accuracy. In the end, the ellipsometric measurements are used as references.

## 2. Theoretical Model of Dielectric Constant from SPR Curves

### 2.1. Relation between Dielectric Constants and SPR Curves

Surface plasmon resonance occurs when the wave vector of incident polarized light is matched with that of the surface plasmon wave on a metal thin film. A typical three-layer Kretschmann structure is shown in Figure 1a. The energy of the incident light is transformed into the surface plasmon wave while the reflective resonance absorption peak is formed. The reflective intensity varies continuously with the incident angle, as shown in Figure 1b.

The SPR curve has three key parameters: the resonance angle (*θ_SPR_*), full width at half maximum (FWHM) (*W_θ_*), and minimum reflectivity (*R_min_*). The complex wave vectors *K* and *K_x_* can be obtained from Kx(θ)=Re(K) at
(1)θ=θSPR=arcsin[Re(K)c/n].

The reflectivity is calculated from the Fresnel equation:(2)$$R={\left|\frac{r_{01}+r_{12}\exp(2ik_{z1}d_{1})}{1+r_{01}r_{12}\exp(2ik_{z1}d_{1})}\right|}^{2}.$$

The amplitude reflective coefficient (*r_mn_*) can be calculated as:(3)$$r_{mn}=\frac{\varepsilon_{m}k_{zn}-\varepsilon_{n}k_{zm}}{\varepsilon_{m}k_{zn}+\varepsilon_{n}k_{zm}}(m=0,1\;n=1,2)$$
(4)$$k_{zi}=\sqrt{{\left(\frac{\omega }{c}\right)}^{2}\varepsilon_{i}-k_{x0}^{2}}\;(i=0,\;1,\;2)$$
where the subscripts 0, 1, and 2 designate the parameters of prism, metal, and air, respectively; *r_mn_* is the amplitude reflectivity of the prism–metal and metal–air interfaces given by the Fresnel formula relating to p-polarization; *k_x0_* is the component of the incident wave-vector parallel to the interface between the prism and metal; *n* is the refractive index of the prism; *ω* is the frequency of the incident light, *c* is the velocity of light in the air; *θ* is the incident angle of the light; *ε_i_* and *k_zi_* are the dielectric constants and the wave-vector components perpendicular to the interface in medium *j*, respectively; and *d* is the thickness of the metal film.

The FWHM and minimum reflectivity are given by:(5)Wθ=2Im(K)cos(θSPR)c/nω
(6)$$R_{\min}=1-4\eta /{\left(1+\eta \right)}^{2}$$
(7)η=Im(K0)/Im(KR)
where *K^0^* is the complex wave vector of the SPR on the metal-vacuum interface in the absence of the prism and *K^R^* is the perturbation to *K^0^* in the presence of the prism. The expanded form of *K^R^* and *K^0^* can be calculated as:(8)$$K^{0}={\left(\frac{\varepsilon_{1}\varepsilon_{2}}{\varepsilon_{1}+\varepsilon_{2}}\right)}^{1/2}\frac{\omega }{c}={\left(\frac{\varepsilon_{1}^{\prime }\varepsilon_{2}}{\varepsilon_{1}^{\prime }+\varepsilon_{2}}\right)}^{1/2}\frac{\omega }{c}+i{\left(\frac{\varepsilon_{1}^{\prime }\varepsilon_{2}}{\varepsilon_{1}^{\prime }+\varepsilon_{2}}\right)}^{1/2}\frac{\varepsilon_{1}^{\prime \prime }\varepsilon_{2}}{2\varepsilon_{1}^{\prime }(\varepsilon_{1}^{\prime }+\varepsilon_{2})}\frac{\omega }{c}$$
(9)$$K^{R}=\frac{\omega }{c}(-r_{01})K=K^{0}(\frac{2}{\varepsilon_{1}-\varepsilon_{2}}){\left(\frac{\varepsilon_{1}\varepsilon_{2}}{\varepsilon_{1}+\varepsilon_{2}}\right)}^{3/2}\times \exp\left[i\frac{4\pi d}{\lambda }\frac{\varepsilon_{1}}{{\left(\varepsilon_{1}+\varepsilon_{2}\right)}^{1/2}}\right].$$

From Equation (1)–(9), we can determine the dielectric constant and thickness of the metal film from the resonance angle, full width at half maximum (FWHM), and minimum reflectivity of the measured surface plasmon resonance sensor (SPR) results. The resonance angle and minimum reflectivity can be obtained directly from the lowest point of the SPR curve. The FWHM needs to be manually calculated using the highest and lowest points of the curve. From the theoretical model, the dielectric constants of the silver film at 432.8 and 632.8 nm are  ε=−5.19+i0.28 and ε=−16.32+i0.54, which are consistent with the experimental results (the dielectric constants of the silver film at 432.8 and 632.8 nm are  ε=−5.25+i0.32 and ε=−16.72+i1.66, respectively). Above all, the theoretical prediction of the dielectric constant is accurate, especially in the real part.

### 2.2. Derivation of the Dielectric Constant from SPR Curves

The derivation of the dielectric constant and thickness includes the following steps (Figure 2):
(1)Substitute the measured *θ_SPR_* into Equation (1) and calculate Re(*K*);(2)Use Re(KR)≫Re(K0) to set Re(K)=Re(K0) and determine $\varepsilon_{1}^{\prime }$ using the real part of Equation (8);(3)Determine [Im(K0)+Im(KR)]=Im(K) by substituting the measured θSPR and Wθ into Equation (5);(4)Determine Im(K0)/Im(KR) by substituting the measured Rmin into Equation (6) and calculate Im(K0) and Im(KR) separately;(5)Determine $\varepsilon_{1}^{\prime \prime }$ by substituting $\varepsilon_{1}^{\prime }$ and Im(K^0^) into the imaginary part of Equation (8);(6)Determine *d* by substituting *θ_SPR_*, $\varepsilon_{1}^{\prime \prime }$, $\varepsilon_{1}^{\prime }$, and Im(K^R^) into the imaginary part of Equation (9).

We obtain two solutions of Im(K0)/Im(KR) in step 4, the two sets of solutions of *ε_1_* and *d* at a specific wavelength, then we repeat steps 1 to 4 with a series of wavelengths. Although *ε_1_* varies with the wavelength, *d* is constant. The average of dielectric constants at the same thickness approaches the optimal result. The major error of the above derivation process comes from the home location accuracy of the motorized continuous rotation stage in the SPR system. The home location accuracy of ±0.2° brings a maximal dielectric constant deviation of 10%.

## 3. Film Preparation and Measurements

In our SPR sensing system, both the prism and the metal deposited substrate was made of N-BK7 glass (*n_0_* = 1.516). The substrate (20 mm × 20 mm × 0.2 mm) was ultrasonically cleaned with ultrapure water, acetone, and pure ethanol for 20 min successively and blow dried with dry nitrogen. Then, a silver film was deposited in a magnetron sputtering coater with a uniformly rotating substrate of 1 Å/s and an argon vacuum degree of 3 × 10^−4^ Pa. In order to investigate the effects of film thickness on the dielectric constants, gold and silver nano-films of 35, 45, and 55 nm were deposited under the same conditions. Considering the oxidation of silver, the ellipsometric and SPR experiments of one film were completed within 30 min after taking the film out of the vacuum chamber.

An angle modulated SPR sensing system with multiple wavelength light sources is shown in Figure 3. A supercontinuum laser source (SC-Pro, YSL) with an output spectrum of 400–2400 nm and a spectral energy density over 1 mW/nm in the visible region was used. After passing through an acousto-optical tunable filter (430–1450 nm), a 2 nm linewidth output arrived at an optical collimator (consists of a self-focusing lens and a polarized filter) before entering the sensing prism. The prism was mounted on a motorized continuous rotation stage (PRMTZ7, Thorlabs) with a repeatable incremental motion (Min) of 0.04° and a percentage accuracy of 0.1%, which enables high-precision continuous angle adjustment. A linear CCD camera received and recorded the reflective signal of the SPR sensor with 5% sensitivity non-uniformity, which was then processed by a computer.

Figure 4 plots the experimental SPR curves of gold and silver films at 45 nm. The parameters of the curves change clearly with the wavelength. The influence of wavelength and thickness on the dielectric constants is discussed in Section 4.

The ellipsometer (T-Solar, J.A. Woollam) has a halogen source in the spectrum of 245–1700 nm with the incident angle range from 45 to 65° and interval of 10°. Two modes, transmission and reflection, are involved for analysis. The referenced experiments on the same silver and gold films in Section 3 were conducted using an ellipsometer.

## 4. Results and Discussions

### 4.1. Experimental Results of SPR Sensors and Ellipsometer

We prepared six gold and silver nano-film samples with three thicknesses and conducted a set of nine measurements of the dielectric constants for each film sample at a wavelength step of 50 nm between 500 and 900 nm using both the proposed SPR system and the ellipsometer. In order to obtain the dielectric constants, we extracted the resonance angle, FWHM, and minimum reflectivity from the SPR curves of different thicknesses using Chen’s model, as shown in Section 2.2 [11]. Both the SPR (triangle marked line) and ellipsometric (solid line) results of the silver nano-films are shown in Figure 5, as well as the most used results of Christy’s model (solid diamond marked line) and the L-D model (hollow diamond marked line). Figure 5a,c,e are the results of the real part, and Figure 5b,d,f are the results of the imaginary part of the dielectric constant. As great uncertainty in the imaginary part exists in Christy’s dielectric constant, we estimated and plotted the measurement uncertainty range with two blue diamond marked lines according to the work in [4]. However, the real part of the dielectric constant is consistent and matched in trend for all six samples. The SPR results were always greater than that of the ellipsometer, which was clearly shown in the results of the 35 nm films. Both results of the imaginary part fell consistently in the upper half of Christy’s uncertainty range; furthermore, the SPR results were located right at the upper boundary.

The experimental results of the dielectric constants of gold nano-film are shown in Figure 6, in which Figure 6a,c,e is the results of the real part, and Figure 6b,d,f is the results of the imaginary part. Similar to the silver films, the real parts of the SPR experiments are consistent with the L-D model and vary with the film thickness. For the imaginary parts, the SPR results are more consistent with Christy’s results, falling in the upper half of the uncertainty range; however, the ellipsometer results fell outside the uncertainty range between 400 and 800 nm.

### 4.2. Effect of Thickness on Dielectric Constant

We extracted three specific parameters, the resonance angle, FWHM, and minimum reflectivity from the SPR curves, to compare the responses to different film thickness as shown in Figure 7. The thickness effects of the above three parameters on silver and gold films were very similar. The resonance angle decreased with increasing wavelength first then slowed down after 700 nm, as shown in Figure 7a,d. Although the variation trend of the resonance angles of all samples were the same, the difference between the films of 45 nm to 55 nm was closer than to 35 nm. The same downward trend of all six samples in FWHM can be found in Figure 7b,e. The minimum reflectivity decreases first then raise up with increasing wavelengths before becoming stable. The reflectivity of silver and gold nano-films decreases first and then increases with thickness, indicating an optimum thickness of the SPR resonance.

Figure 8 plots the SPR and ellipsometer measurement results of the thickness effects on both the real and imaginary parts of the dielectric constants of gold and silver films. The curves of SPR and the ellipsometer measurements exhibited slight differences under the same thickness, but demonstrated the same trend. The change in the ellipsometer results with the thickness was noticeably smaller than the SPR results surface plasmon resonance (SPR).

Except for the imaginary part results of gold nano-film, which are consistent with the SPR experiment, the changes of other parameters were irregular, which showed that the influence of the systematic error of the ellipsometer instrument error is greater than that of the film thickness on the dielectric constant. The SPR experimental results prove that the value of the dielectric constant decreases when the thickness increases. The dielectric constant changes less with film thickness when the film thickens, which indicates that the dielectric constant of the film tends to the bulk of the material over 30 nm. Correspondingly, a greater change in the dielectric constant value is exhibited in the thinner films. As the SPR effect becomes very weak after the film thickness reduces to 30 nm or less, the film thickness effects on the dielectric constant are unable to be effectively investigated with the SPR method.

The dielectric constant directly obtained from the SPR method reduced the relative error up to 80% compared with the typical optical methods that require the optical constants *n* and *k*. The backstepping algorithm calculates the optical constants $n=\sqrt{1/2\left(\sqrt{\varepsilon_{1}^{\prime }+{\varepsilon_{1}^{\prime \prime }}_{1}}+\varepsilon_{1}^{\prime }\right)}$ and $k=\varepsilon_{1}^{\prime \prime }/2n$ from the dielectric constant results of SPR, as shown in Figure 9. It is apparent from Figure 9 that the optical constant varies with the thickness of the films. The thickness decrease accelerates the increase of the refractive index. Conversely, the extinction coefficient has an accelerated decrease with the thickness. The SPR experimental results of the optical constants of the metal thin films are close to the predictions of C. Reale [20].

### 4.3. Systematical Error

To understand the experimental deviation from the classical models, the derivation processes of the dielectric constant was theoretically simulated using the previously mentioned two methods for the noble metal nanofilms. In the Drude free-electron theory, the dielectric constant is expressed as:(10)$$\varepsilon_{1}(\omega )=1-\frac{\omega_{p}^{2}}{\omega (\omega +i/\tau )}$$
(11)$$\omega_{p}^{2}=\frac{4\pi Ne^{2}}{m_{0}}$$
where *N* and *m_0_* are the density and effective mass of the conduction electrons, and *τ* is the relaxation time. Separating the real part from the imaginary part of *ε_1._*
(12)$$\varepsilon_{1}^{\prime }=1-\frac{\omega_{p}^{2}\tau^{2}}{1+\omega^{2}\tau^{2}}$$
(13)$$\varepsilon_{1}^{\prime \prime }=\frac{\omega_{p}^{2}\tau }{\omega (1+\omega^{2}\tau^{2})}$$

1/τ is found in the visible to NIR region for the relaxation time of the noble metals being on the order of 10^−15^ s. The Drude model obtains the real and imaginary parts of the dielectric constant as:(14)$$\varepsilon_{1}^{\prime }≅1-\frac{\omega_{p}^{2}}{\omega^{2}}=1-\frac{\lambda^{2}}{\lambda_{p}^{2}}$$
(15)$$\varepsilon_{1}^{\prime \prime }≅\frac{\omega_{p}^{2}}{\omega^{3}\tau }=\frac{\lambda^{3}}{\lambda_{p}^{2}}\tau^{\prime }$$

In their prior research, Christy et al. measured the refractive index and the extinction coefficient [4] based on the complex permittivity $\varepsilon_{1}={\tilde{n}}^{2}$, where $\tilde{n}=n+ik$, and measured the real and imaginary values of the dielectric constant using the equations, $\varepsilon_{1}^{\prime }=n^{2}-k^{2}$ and $\varepsilon_{1}^{\prime \prime }=2nk$. As the refractive index is small, less than 0.1 (the minimum of 0.03) in the visible to NIR region, an instrumental error of 0.02 leads to a large uncertainty in the imaginary part of the dielectric constant. Moreover, the relative error is only 0.007 for the extinction coefficient that results from the instrumental error.

In the SPR experiments, the major systematic error is the home location accuracy of ± 0.2° of the motorized continuous rotation stage, which results in a relative error of 3.27% for the real part and a relative error of approximately 6.95% for the imaginary part. The error introduced by the sensing method is 80% less than the other optical technologies interpreted from the refractive and extinction coefficient values. The finding of the SPR measurements that the imaginary part of the dielectric constant of silver and gold nano-films always falls in the upper half of the data uncertainty region of Christy’s results, indicates that the refractive index is always less than the true value, with a negative error in the measurement from Christy’s model. The significant thickness effects on the parameters of the SPR curves weaken as the film thickness increases.

## 5. Conclusions

We investigated the dielectric constant of noble metal nano-films using the SPR method. This method has a higher precision than the traditional optical methods, and reduces about 80% of the relative error of the imaginary part to 6.95%. Compared with the results from Christy’s method, the imaginary part of the experiment was always located within the upper boundary of the uncertainty range in the 500–900 nm spectral region. The experiments with various film thicknesses less than 50 nm showed that the dielectric constants and optical constants of the noble metal film are affected by the film thickness. Specifically, the values of the dielectric constant and refractive index increase with a decrease in the thickness. However, the change in the extinction coefficient is the opposite to the change in the refractive index. Simultaneously, the changes in all parameters become more severe as the films thin.

## Figures and Tables

**Figure 1 sensors-20-01505-f001:**
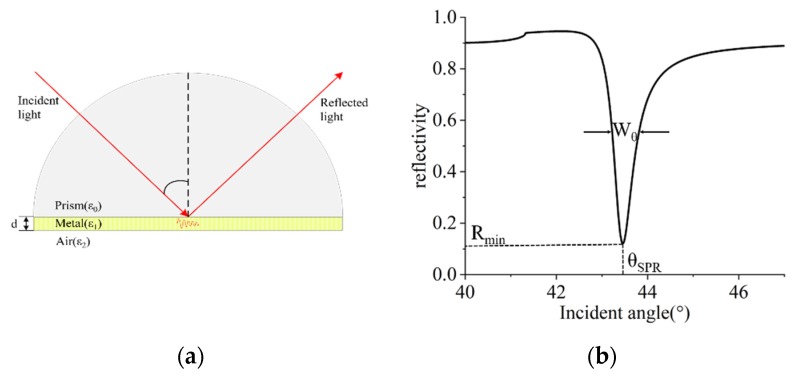
(**a**). A prism–metal–air Kretschmann configuration; (**b**) the reflectivity (*R*) versus the incident angle θ.

**Figure 2 sensors-20-01505-f002:**
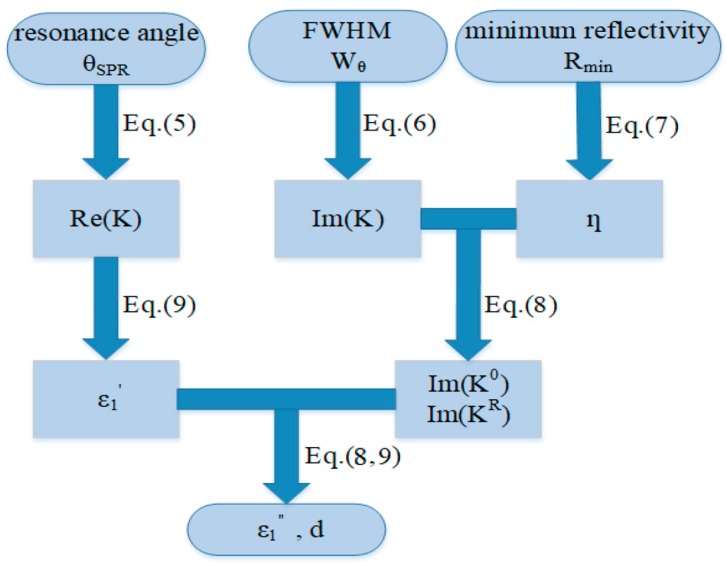
Flow chart for the dielectric constant calculation. Full width at half maximum (FWHM).

**Figure 3 sensors-20-01505-f003:**
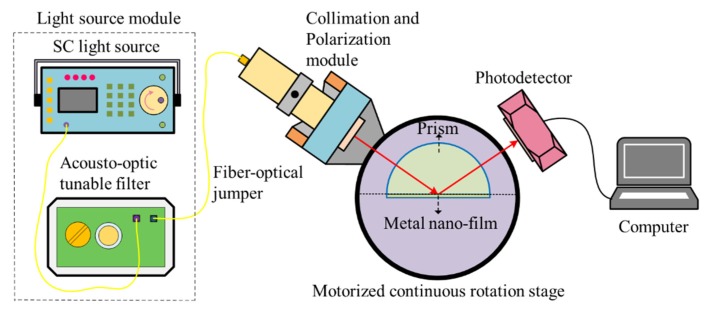
The experimental setup of the angle modulated surface plasmon resonance (SPR) system.

**Figure 4 sensors-20-01505-f004:**
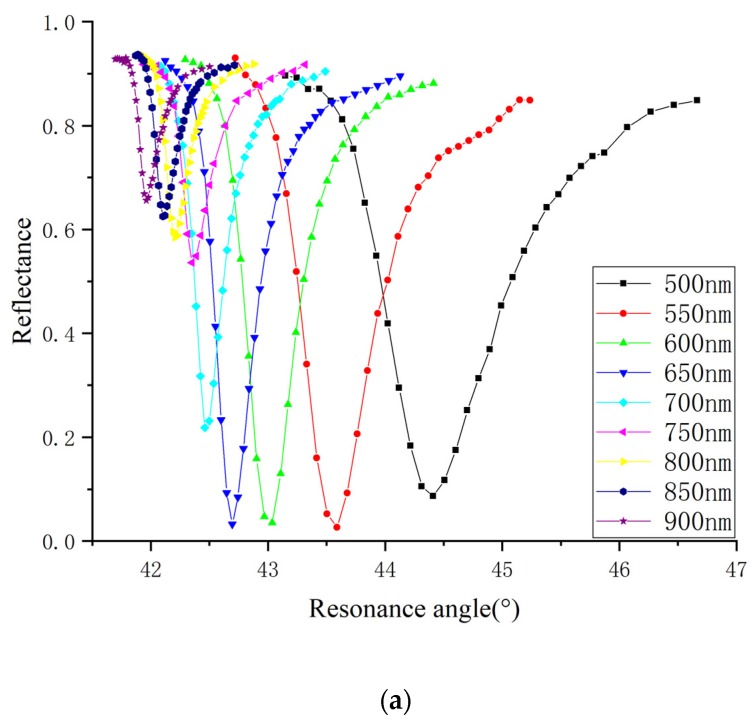
The shift in reflectance with the resonance angle under the different wavelengths: (**a**) 45 nm silver nano-film; (**b**) 45 nm gold nano-film.

**Figure 5 sensors-20-01505-f005:**
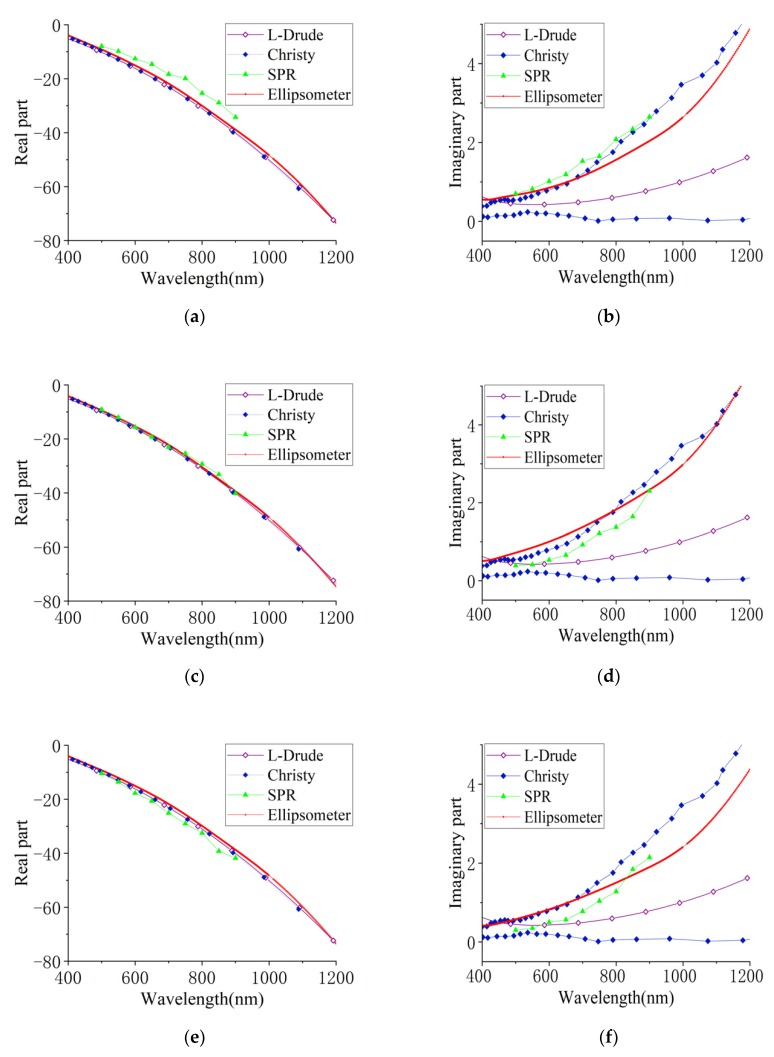
The dielectric constant theoretical fitting and experimental measurement results of (**a,b**), 45 nm (**c,d**), and 55 nm (**e,f**) silver films.

**Figure 6 sensors-20-01505-f006:**
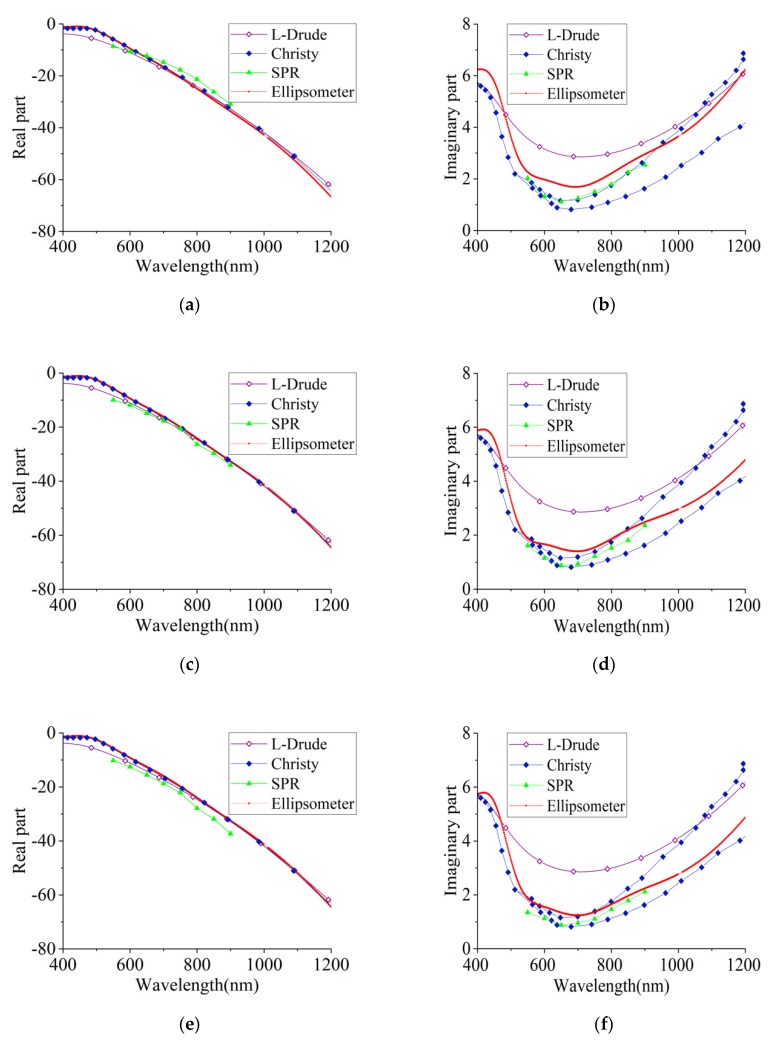
Dielectric constant theoretical fitting and experimental measurement results of the 35 nm (**a,b**), 45 nm (**c,d**), and 55 nm (**e,f**) gold films.

**Figure 7 sensors-20-01505-f007:**
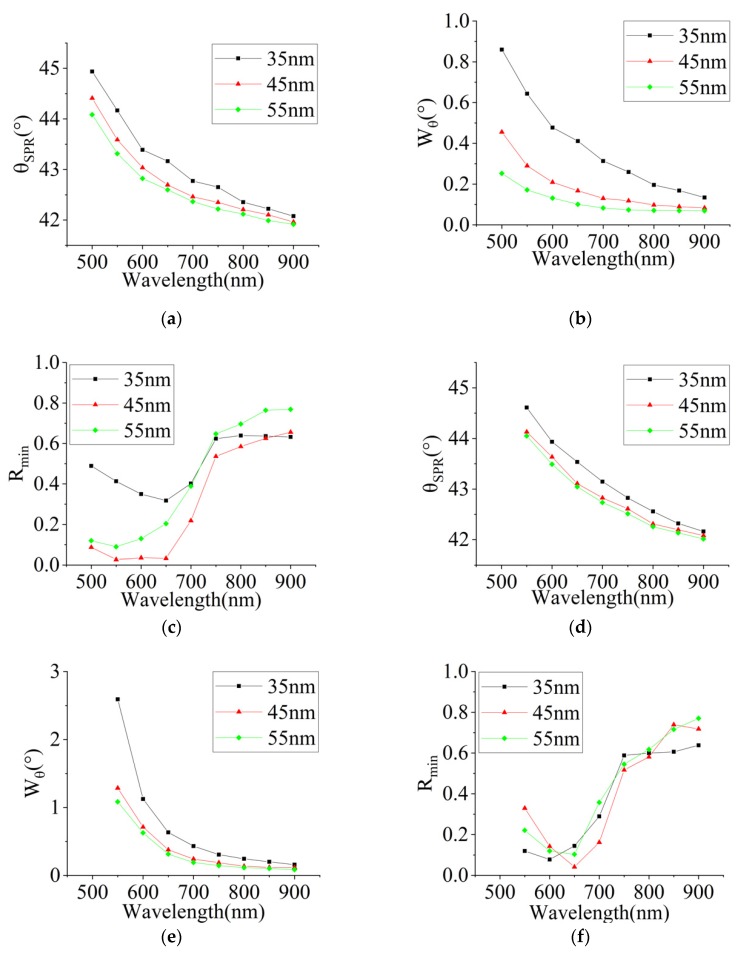
The minimum reflectivity, resonance angle, and FWHM measurement results of silver (**a–c**) and gold (**d–f**) films.

**Figure 8 sensors-20-01505-f008:**
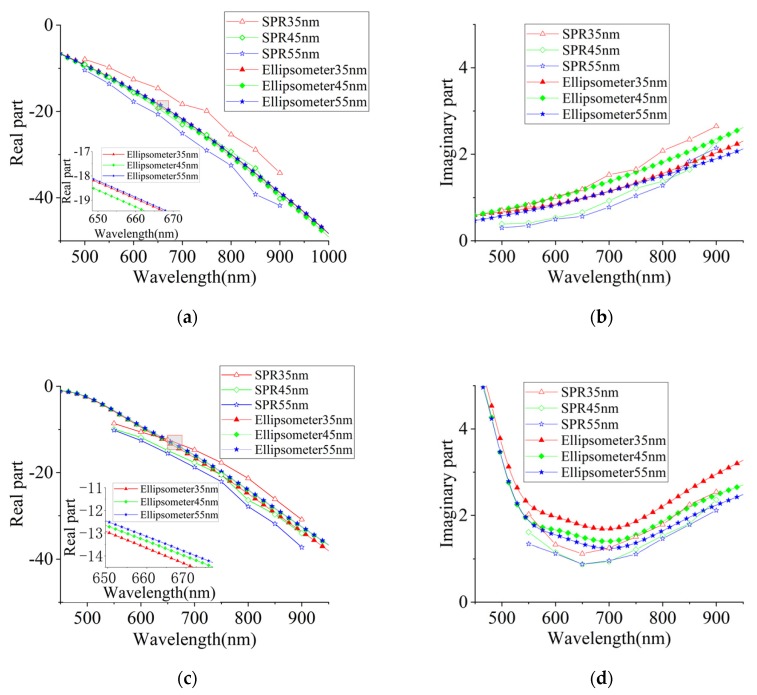
The dielectric constant measurement results of the SPR and Ellipsometer: (**a**) the real part of the silver films; (**b**) the imaginary part of the silver films; (**c**) the real part of the gold films; and (**d**) the imaginary part of the gold films.

**Figure 9 sensors-20-01505-f009:**
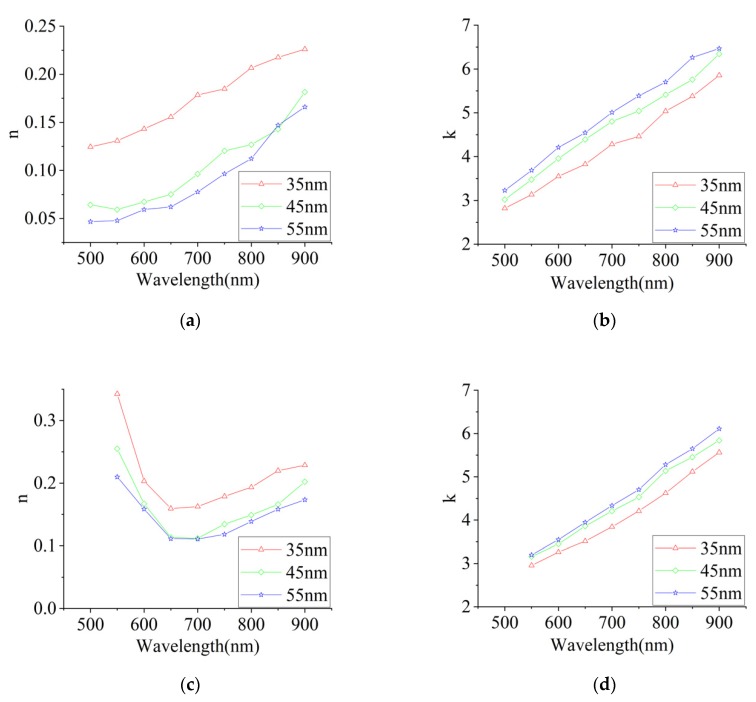
The refractive index and extinction coefficient spectrums of silver (**a,b**) and gold (**c,d**) films for different thicknesses.

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
