# Peer review of "Experimental Investigation of the Dielectric Constants of Thin Noble Metallic Films Using a Surface Plasmon Resonance Sensor"

_sensors, 2020, doi:10.3390/s20051505_

Round 1
Reviewer 1 Report
The manuscript reports about a method for measuring the optical properties of thin metal films, alternative to the standard ellipsometry approach. The topic is interesting, the research is well presented and the manuscript deserves publication. I only suggest few minor corrections.
- Reference 2 is probably located in the wrong place in the text. I don't feel it is a reference to support the standard textbook Lorentz-Drude model.
- The introduction is too generic. The manuscript would be significantly improved with a more detailed introduction about the state of the art methods. As it is now, it is just a list of references without a sufficient discussion about the characteristics of the mentioned methods.
- Section 2.1 reports in detail a theoretical model for the SPR already reported by other (correctly cited) authors. It would be interesting if the physical approximations (and limitations) underlying this model could be outlined by the authors. This could be of great help for other authors interested in applying the model to other similar situations.
- Figures (4 to 9) are too small and there is full of space to enlarge them.
- Figure captions (4 to 9) must be improved. the reader should be able to get the focus of the results without recurring ho the description on the body text.
- Line 196: the word "show" is misused.
- Description of figure 8 (lines 187-192) should be improved, I can't see a relation between the description presented and the figure itself (maybe because it is too small?)
Reviewer 2 Report
Report on the manuscript sensors-726322:“Experimental investigation of the dielectric constants of thin noble metallic films using surface plasmon resonance sensor” by Longbiao Tao et alii
The work describes a multi-wavelength angle-modulated SPR sensing technology to get high precision optical constants of the noble metal film. Since the spectral sensitivity and amplitude of SPR curves depend on the thickness and the dielectric constant of the sensing layer, the authors carried out (compared with standard ellipsometric data and tables in literature) a set of measurements with the aim to evaluate precisely real and imaginary parts of silver and gold nano-films. The work technically sounds and is well balanced in terms of theoretical and experimental findings making it very interesting. For these reasons, I suggest to publish the present manuscript in Sensors MDPI journal. In doing that, I warmly suggest to the authors to clarify/adjust the following minor issues:
1) In the Abstract, the sentence: “The extremely low value about 0.04 in visible to near-infrared (NIR) results in a large relative error reaching 50% or even more in refractive index, thus leading to a large uncertainty of the imaginary part of the dielectric constants” has to be rephrased. It is hard to understand. Please check and adjust the first part of the Abstract.
2) In the Abstract, the sentence : ” On the other hand, surface plasmon resonance sensors (SPRs), which uses tens of nanometer-thick noble metal film as the sensing layer, shows ultra-high sensitivity (reaching 10-8 RIU) in this spectral range” has to be properly quoted in literature. Please add a reference to such a sensitivity for SPR.
3) In Section 2, there is missing information. How did you extract the resonance angle, FWHM, and minimum of reflectivity from SPR curves? What did you use as a fitting function for the SPR curves?
4) I would like to advise the authors to find a native English speaker to proofread the manuscript.
